



# Tipping point analysis of ocean acoustic noise

**Valerie N. Livina**[1], **Albert Brouwer**[2], **Peter Harris**[1], **Lian Wang**[1], **Kostas Sotirakopoulos**[1], **and Stephen Robinson**[1]

[1]National Physical Laboratory, Teddington, Middlesex TW11 0LW, UK
[2]Preparatory Commission of the Comprehensive Nuclear-Test-Ban Treaty Organization, Vienna, Austria

**Correspondence:** Valerie N. Livina (valerie.livina@npl.co.uk)

**Abstract.** TS1 TS2 We apply tipping point analysis to a large record of ocean acoustic data to identify the main components of the acoustic dynamical system and study possible bifurcations and transitions of the system. The analysis is based on a statistical physics framework with stochastic modelling, where we represent the observed data as a composition of deterministic and stochastic components estimated from the data using time series techniques. We analyse long-term and seasonal trends, system states and acoustic fluctuations to reconstruct a one-dimensional stochastic equation to approximate the acoustic dynamical system. We apply potential analysis to acoustic fluctuations and detect several changes in the system states in the past 14 years. These are most likely caused by climatic phenomena. We analyse trends in sound pressure level within different frequency bands and hypothesize a possible anthropogenic impact on the acoustic environment. The tipping point analysis framework provides insight into the structure of the acoustic data and helps identify its dynamic phenomena, correctly reproducing the probability distribution and scaling properties (power-law correlations) of the time series.

## 1 Introduction

The Preparatory Commission of the Comprehensive Nuclear-Test-Ban Treaty Organization (CTBTO) has established a global network of underwater hydrophones as a part of its hydroacoustic observations (others being seismic, infrasound, and radionuclide), with the goal of continuous monitoring for possible nuclear explosions (CTBTO, 2013). The CTBTO database provides several unique and large oceanic acoustic records, covering more than 10 years of continuous recording with a high temporal resolution of 250 Hz. In this article, we study the records of the hydrophone H01W1 at the Cape Leeuwin station. The hydrophone is located at a depth of about 1 km off the southwest shore of Australia. We apply tipping point analysis and identify the main components of this acoustic dynamical system, which we then model, with reconstruction of the probability distribution and scaling properties (power-law correlations) of the observed data. Both the probability distribution and scaling properties are important for ensuring that the model correctly represents the observed data, because probability distribution characterizes the range and frequency of time series values, while scaling properties characterize their temporal arrangement (Kantelhardt et al., 2002; Livina et al., 2013).

Tipping points in climatic subsystems have become a widely publicized topic of high societal interest related to climate change; see, for example, Lenton et al. (2008). Applications of tipping point analysis have been found in geophysics (Lenton et al., 2009, 2012a, b; Livina and Lenton, 2007; Livina et al., 2010, 2011, 2012, 2013; Cimatoribus et al., 2013), statistical physics (Vaz Martins et al., 2010; Livina et al., 2013), ecology (Dakos et al., 2012), and structure health monitoring (Livina et al., 2014; Perry et al., 2016).

A stochastic model combining deterministic and stochastic components is a powerful yet simple tool for modelling time series of real-world dynamical systems. Given a one-dimensional trajectory of a dynamical system (the recorded time series), the system dynamics can be modelled by the stochastic equation with state variable $z$ and time $t$:

$$\dot{z} = D(z, t) + S(z, t), \tag{1}$$

where $\dot{z}$ is the time derivative of the system variable $z(t)$, and $D$ and $S$ are deterministic and stochastic components, respectively. Component $D(z, t)$ may be stationary or dynam-

ically changing (for instance, containing long-term and/or periodic trends). Many geophysical variables, which follow seasonal variability, can be approximated by a stochastic model

$$z(t) = T(t) + A(t)\cos(2\pi\phi(t)) + \Phi(t), \tag{2}$$

where the trend $T(t)$ is a real-valued function, such as a straight-line function of $t$, the second term models seasonal variability, and $\Phi(t)$ is a stationary random process. As an example, $\Phi(t)$ can be Gaussian white noise or a continuous autoregressive moving average random process of order $(p, q)$. Similarly, De Livera et al. (2011) used a trigonometric Box–Cox transform with ARMA errors and seasonal components.

The probability distribution of the trajectory (time series) of such a system, however complex it may be, can in the majority of cases be approximated using a so-called system potential in the form of a polynomial of even order. Tipping points can be identified in terms of the variability of the underlying system potential $U(z, t)$, which defines (if it exists) the deterministic term in Eq. (1): $D(z, t) = -U'(z, t)$. If the structure of the potential (the number of potential wells) changes, the tipping point is a bifurcation. If the potential structure remains the same, while the trajectory of the system samples various states, such a tipping point is transitional (Livina et al., 2011). The stochastic component, in the simplest case, may be Gaussian white or red noise, with possible multifractality and other nonlinear properties. The tipping point methodology is currently based on the techniques of degenerate fingerprinting and potential analysis, which are described below.

## 2 Methodology

The tipping point analysis consists of the following three stages: (1) anticipating (pre-tipping, or analysis of early-warning signals), (2) detecting (tipping), and (3) forecasting (post-tipping).

*Anticipating tipping points* (pre-tipping) is based on the effect of slowing down of the dynamics of the system prior to critical behaviour. When a system state becomes unstable and starts a transition to another state, the response to small perturbations becomes slower. This "critical slowing down" can be detected as increasing autocorrelations (ACF) in the time series (Held and Kleinen, 2004). Alternatively, the short-range scaling exponent of detrended fluctuation analysis (DFA) (Peng et al., 1994) may be monitored (up to 100 units, which in the case, for example, of daily data correspond to 3.5 months; see Livina and Lenton, 2007). The lag-1 autocorrelation is calculated in sliding windows of fixed length (conventionally, half of the series length) or variable length (for uncertainty estimation) along the time series, which produces a curve of an early warning indicator. This indicator describes the structural dynamics of the time series.

If the curve of the indicator remains flat and stable, the time series does not experience a critical change (whether bifurcational or transitional). If the indicator rises to a critical value of 1 (the monotonic trend can be estimated, for instance, using Kendall rank correlation), it provides a warning of critical behaviour.

Lag-1 autocorrelation is estimated by fitting an autoregressive process of order 1 (AR1):

$$z_{t+1} = cz_t + \sigma\eta_t, \tag{3}$$

where $\eta$ is a Gaussian white noise process of unit variance, $\sigma$ is the noise level, and $c = e^{-\kappa\Delta t}$ is the "ACF indicator" with $\kappa$ the decay rate of perturbations. Then, $c \to 1$ as $\kappa \to 0$ when a tipping point is approached. In addition, the DFA method utilizes built-in detrending of a chosen polynomial order, which allows one to distinguish transitions and bifurcations in the early-warning signals. This can be done by comparing several early-warning indicators, with and without detrending data in sliding windows (Livina et al., 2012). The paper Livina and Lenton (2007) provided the first application of the DFA-based early-warning indicator to the paleotemperature record with detected transition using both ACF and DFA indicators.

*Detecting* and *forecasting* of a tipping point is performed using dynamical potential analysis. The technique detects a bifurcation in a time series and the time when it happens, which is illustrated in a novel plot mapping by colour the potential dynamics of the system (Livina et al., 2010, 2011). The dynamics of the tipping point is forecast using extrapolation of the dynamically derived Chebyshev coefficients of the approximation to the probability density function of the system trajectory (Livina et al., 2013).

For the purposes of potential analysis, the dynamics of the system is approximated by a potential stochastic model with a polynomial $U$ (which, in general, may depend on both state variable $z$ and time)

$$\dot{z} = -U'(z, t) + \sigma\eta, \tag{4}$$

where $\dot{z}$ is the time derivative of the system variable $z(t)$, $\eta$ is Gaussian white noise of unit variance, and $\sigma$ is the noise level. In the case of a double-well potential, $U$ can be described by a polynomial of fourth order (assuming its quasi-stationarity, with dependence on the state variable $z$ only):

$$U(z) = a_4 z^4 + a_3 z^3 + a_2 z^2 + a_1 z. \tag{5}$$

The Fokker–Planck equation for the probability density function $p(z, t)$,

$$\delta_t p(z, t) = \delta_z \left[ U'(z) p(z, t) \right] + \frac{1}{2}\sigma^2 \delta_z^2 p(z, t), \tag{6}$$

has a stationary solution given by

$$p(z, t) \sim \exp\left[-2U(z)/\sigma^2\right]. \tag{7}$$

The potential can be reconstructed from time series data of the system using the following relation to the probability density function:

$$U(z) = -\frac{\sigma^2}{2} \log p_d(z), \tag{8}$$

which means that the empirical probability density $p_d$ (kernel distribution) has a number of modes corresponding to the number of wells of the potential.

The structural changes of the potential are often not visible in the time series, yet they may lead to a dramatic evolution of the system. Detecting such changes gives an advantage in understanding of the dynamical system. The potential coloured map (Livina et al., 2010) visualizes bifurcations according to the number of detected system states. It illustrates bifurcations as the change in the colour describing the number of states along all timescales (the $y$ axis shows the length of the sliding window of data for which the number of states is assessed). If no such pattern is observed, there is no bifurcation in the time series.

This stochastic approximation of the system structural dynamics has remarkable accuracy for data subsets of length as short as 400 to 500 data points, demonstrating above 90 % rate of successful detection, as was shown in an experiment with double-well-potential artificial data (Livina et al., 2011). For data subsets of length greater than 1000 data points it correctly detects the structure of the potential with a success rate of over 98 %.

The technique of potential forecasting is based on dynamical propagation of the probability density function of the time series. We employ the coefficients of the Chebyshev polynomial approximation of the empirical probability distribution and extrapolate them in order to forecast the future probability distribution of the data. After reconstruction of the system kernel distribution, a time series is generated using rejection sampling technique, and then the obtained dataset is sorted according to the initial data in order to reconstruct the temporal correlations in the time series. The detailed mathematical description of the potential forecasting technique is given in Livina et al. (2013). The technique has the advantage of reproducing both static properties (probability density) and dynamic properties (scaling exponent, or power-law correlations) of a time series.

## 3  Data

We study the large CTBTO record (2003–2016) of the Cape Leeuwin hydrophone, series H01W1, which is a 250 Hz sampled time series of ocean sound pressure. The raw data represent 3 TB of binary waveforms, which after extraction constitute 95 billion points in the time series. We analyse 1 min averages of sound pressure level (SPL) in five frequency bands (broadband, 10–30, 40–60, 56–70, and 85–105 Hz), of about 7 million points per time series. These data have pronounced

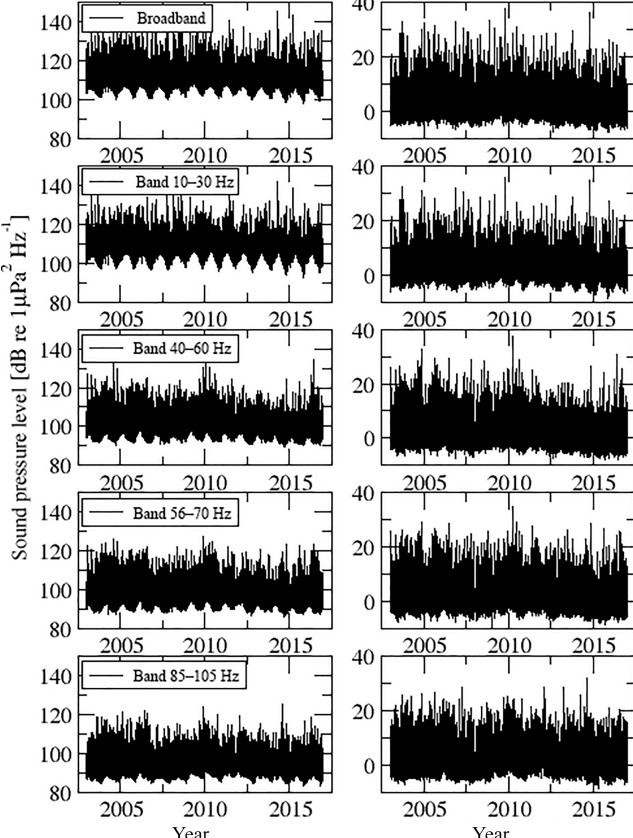

**Figure 1.** CE1 Initial **(a)** and processed **(b)** sounds pressure level data in five frequency bands. Processing included interpolation and deseasonalization. Note that seasonal variability is less pronounced in the higher frequencies of the initial data. At the same time, the records of higher frequencies have declining trend visible by eye.

seasonality and some small gaps, and therefore we perform interpolation and deseasonalization of all five time series, the result of which can be seen in Fig. 1.

The data samples were scaled using their calibration factors (provided by CTBTO), and an inverse filter of the recording system's frequency response was applied to eliminate the effect of the acquisition chain on the frequency response of the recordings. The fast Fourier transform (FFT) of the signal was computed using rectangular windows of 15 000 samples (i.e. 1 min intervals at 250 Hz sampling rate) and the broadband signal was then filtered in five frequency bands (5–115, 10–30, 40–60, 56–70, 85–105 Hz) via selection of the corresponding FFT bins within each frequency band. Then the resulting sound pressure level (SPL) in dB re $1\,\mu\,\text{Pa}^2$ for each frequency band was calculated (Robinson et al., 2014; ISO 18405, 2017). Finally, outliers, i.e. levels greater than 20 dB from the average of the entire time series of SPL values, were removed.

Because of the data gaps, we interpolate the SPL data to achieve equidistant 1 min temporal resolution. We then re-

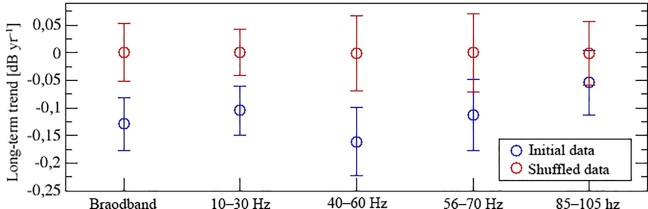

**Figure 2.** Trend estimation in SPL bands (deseasonalized data) using "delete-$d$" jackknife sampling with 1000 subsets with $d = 10\,000$ randomly deleted data points. Blue dots show the slope with corresponding jackknife uncertainties of the least-squares linear regression, whereas red dots show the trend (the absence of it, with zero slopes of linear regression) for the shuffled data, i.e. the data with randomly allocated values of the time series.

move the seasonal periodicity by subtracting the averaged seasonal cycle over the 14 years of observation to obtain the fluctuations

$$z_i = S_i - \bar{S}_i, \tag{9}$$

5 where $S_i$ is the interpolated SPL data, and $\bar{S}_i$ is the mean 1 min interpolated SPL data. The resulting fluctuations are shown in the right column of Fig. 1, for the broadband and four selected sub-bands.

## 4  Results and discussion

10 We analyse the global trends of these five datasets, assuming the simplest linear model in a least-squares regression. To estimate the uncertainty in the trends, we apply the "jackknife" technique (see Efron, 1982; Wu, 1986). We use the "delete-$d$" variation of the method, with random subsampling and 15 numerical implementation reducing the number of required samples, which allows to estimate variance of the trend as

$$v = \frac{r}{dm} \sum_{t=1}^{m} \left( T_{r,s_t} - \frac{1}{m} \sum_{k=1}^{m} T_{r,s_k} \right)^2, \tag{10}$$

where $d$ is the number of the excluded data points in each sample ("delete-$d$") of length $r = n - d$ ($n$ is data length), $T$ 20 are statistics of the trend estimator; see further details in Shao and Tu (1995).

The resulting trends show a small annual decline in SPL for all five datasets, as shown in Fig. 2.

The above trend analysis was applied to deseasonalized 25 fluctuations (SPL broadband). It is interesting that the average annual cycle of the initial broadband data, too, has a declining trend, which is illustrated in Fig. 3.

The origin of the seasonality in the acoustic data from a hydrophone installed at depth is a subject of discussion, 30 because the seasonal ocean temperature fluctuations at the surface would barely influence the sound propagation towards hydrophones. There are various possible mechanisms

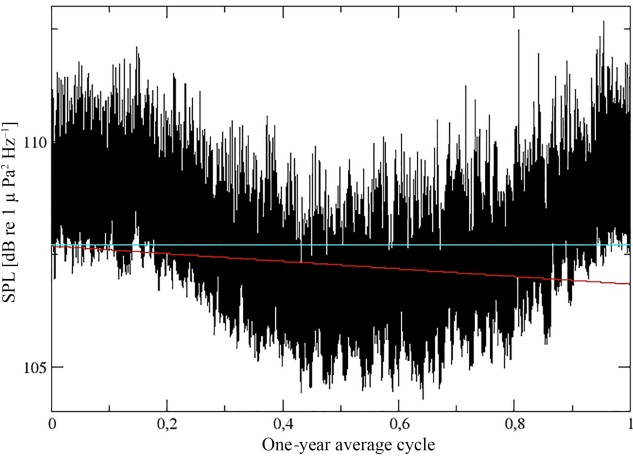

**Figure 3.** Average annual cycle of the SPL broadband, its linear regression line (red) and horizontal line (cyan) for comparison.

through which seasonal variability may manifest in the hydroacoustic data. For example, seasonal variations in shipping frequency, recreational vehicle use, iceberg breakup. 35 Seasonally, there may be a slight warming/cooling in the top few tens of metres of water surface layer, but at the depth of the SOFAR (sound fixing and ranging) channel temperature is stable on a seasonal timescale. Some seasonal effects in the sound record may be originating from iceberg forma- 40 tion as the edges of the Antarctic, as there are slightly faster ice crumbling in the Southern Hemisphere summertime. Furthermore, seasonal variations in whale song are plausible, as well as in fauna migration due to seasonal fluctuations in food supply. 45

We next apply the pre-tipping analysis (early-warning signals) to analyse lag-1 autocorrelations and variance of the broadband SPL record, with estimation of uncertainty. We vary the length of the sliding windows for calculating these indicators between one-fourth and three-fourths of the record 50 length to obtain the averaged curves and standard uncertainties and display the indicator values at the end of each window, as shown in Fig. 5.

The noticeable change at the end of these early-warning indicators may be related to the unusually large El Niño event 55 of 2015–2016. One can see that the variance decline slows down and autocorrelation sharply rises, which means that the increase in memory is not accompanied by increasing amplitude of acoustic fluctuations. Such effects may happen when a dynamical system experiences critical slowing down 60 prior to a bifurcational tipping. As we hypothesize that the El Niño signature may be related to changes in both oceanic dynamics and fauna, the increasing memory in the acoustic data may reflect, for instance, the observation that during the El Niño the Cape Leeuwin current slows down (Feng et al., 65 2003). The slower ocean current introduces more inertia in the dynamical system, and therefore higher temporal memory/autocorrelations.

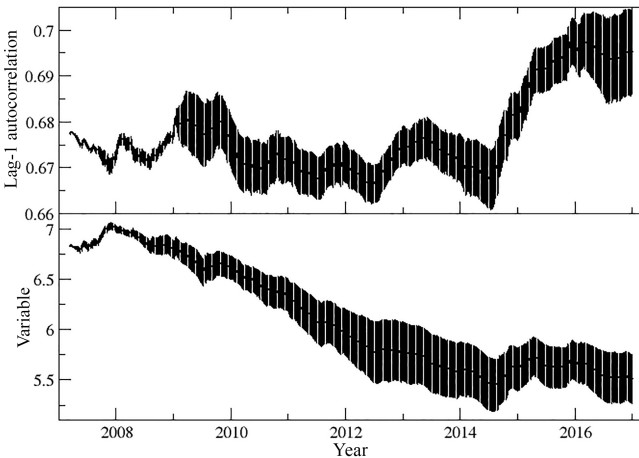

**Figure 4.** Early-warning indicators of the broadband SPL dataset: lag-1 autocorrelation (upper panel) and variance (lower panel), calculated with variable window lengths, from 1/4 to 3/4 of the record length, and corresponding standard uncertainties, displayed at the end of each window. Both indicators demonstrate nonstationary behaviour (increasing autocorrelation and decreasing variance), which denotes long-term development of a possible tipping in future.

Similarly to CTBTO data, the effect of increasing autocorrelation and decreasing variance was earlier observed in bifurcating artificial data changing from white noise to random walk, in Livina et al. (2012). The acoustics dynamics may be undergoing a similar tipping. Note that this analysis of early-warning signals is performed with large enough windows (starting from length of 3 years up to 9 years), which identify large-scale variability, with possible dynamics on the scale of decades ahead.

Further, we apply potential analysis to identify smaller-scale variability, varying the length of the sliding window from 3 days to 1 year. The resulting potential plot is shown in Fig. 5.

El Niño–Southern Oscillation (ENSO) can be monitored using several indices, which are obtained by averaging climatic variables to make the presence of El Niño more visible in the series. We show in Fig. 5 two of them: Southern Oscillation Index (SOI) and Oceanic Niño Index (ONI). SOI is based on the sea level pressure differences between Tahiti and Darwin, Australia. ONI is based on the 3-month running mean of sea-surface temperature anomalies ERSST.v4 SST (Huang et al., 2014) in the Niño 3.4 region (NOAA SOI). Negative SOI (positive ONI) corresponds to El Niño events, characterized by warm SST in the eastern and central tropical Pacific (Trenberth and Caron, 2000).

We calculate, for easier comparison of El Niño indices and potential analysis, two binary indices derived from the ONI and from the single level of the potential plot at the scale 0.5 years. The bars in the bottom panel of Fig. 5 show the occurrence of El Niño events in the ONI (which is less noisy than SOI), and at the same time we plot a binary in-

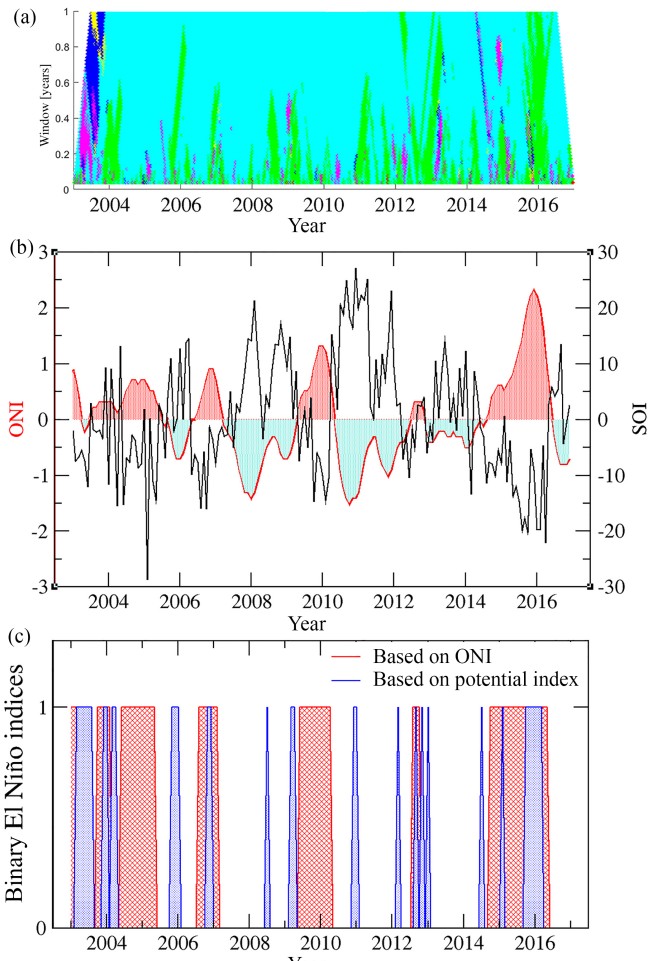

**Figure 5. (a)** Potential analysis plot of the broadband SPL data, with varying window length (*y* axis) from 1 day to 1 year. The colours denote the number of detected potential states: green – two; cyan – three. Specks of magenta denote very short periods of a higher number of states, which correspond to highly variable (possibly non-potential) subsets of data of small size. **(b)** ENSO indices ONI and SOI, known to be anti-correlated, which indicate several ENSO events (El Niño and La Niña). These can also be seen in the potential analysis plot. ONI positive and negative values (roughly corresponding to El Niño and La Niña) are shaded by light red and light blue respectively, for better comparison with the indices in the lower panel. **(c)** Binary indices derived from the ONI and potential plot (at the level 0.5 year of *y*-scale in the top panel) which have values 1 when there is an El Niño (in the case of ONI-based binary index) or anomalous potential state (in the case of the potential binary index).

dex showing periods when the system potential does not follow its "regular" three-well-potential pattern; these two indices have agreement in several periods corresponding to the known El Niño events (2003, 2004, 2006, 2009, 2015–2016), which illustrates our hypothesis of the El Niño signature in the acoustic data.

The vertical span of the features of the potential plot (the specks of different colours) corresponds to the timescale of

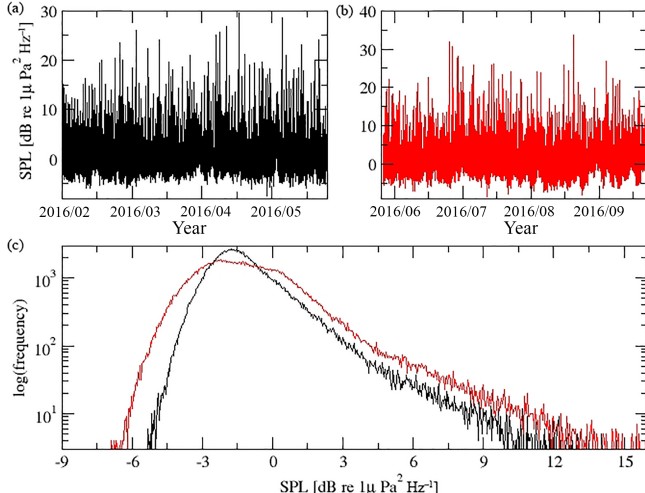

**Figure 6.** The difference between two-well-potential (first half of the year 2016, black curve) and three-well-potential (second half of 2016, red curve) subsets of the broadband SPL record in the two upper panels. In the lower panel, the build-up of the new potential well can be seen in the red histogram, where the main mode becomes broader and starts building two new modes around SPL values $-3$ and $0$ dB re 1 Pa$^2$ (deseasonalized SPL data).

the change, i.e. the size of the time window, within which the change has been detected. As El Niño is a seasonal phenomenon (except the unusually long event of 2015–2016), most of such specks are located within the window of size 1 year. The large event of 2015–2016, indeed, extends higher than that. To address this timescale, we derived the binary potential index using the detection data at fixed timescale of 0.5 year, at which most El Niño events should be present in the detection statistics.

We do not claim that the potential colour plot could be used for early-warning signals (such as prognosis of El Niño), in this system or in others. Moreover, there may be other factors causing structural changes in the acoustic data, rather than El Niño or La Niña. On the other hand, detection of such changes, indeed, can be useful for other studies that could investigate attribution of structural variability, and here the technique of potential analysis might be very useful.

To understand better what dynamical changes occur in the acoustic fluctuations, it is useful to plot the histograms of the corresponding subsets of data. Figure 6 demonstrates the difference between the two-well-potential (first part of the year 2016) and the three-well-potential (second part of the year 2016) subsets, which correspond to green and cyan areas in the top panel of Fig. 5.

The variability of the potential can be understood as appearance and disappearance of the SPL fluctuations, which are present in the three-well-potential subsets and disappear in sub-periods of two-well-potential dynamics. These periods of change seem to coincide with some of the recent El Niño events, in particular the strong oscillation in 2015–

2016. Since in these short periods data become two-well potential during El Niño, one can hypothesize that the El Niño event reduces acoustic fluctuations events in the tails of the probability distribution (higher and lower values) and intensifies the events in the middle range of values.

It is known (Feng et al., 2003) that the Leeuwin Current is influenced by El Niño, which causes lower temperature and slower current. This causes a number of climatic and environmental changes (including the impact of El Niño on sea level, current transport, and migration of marine animals), and this may affect the acoustic signal. In particular, the local sea bottom slopes near Cape Leeuwin are very steep, with large underwater peaks (see Fig. 4 in Feng et al., 2003), which may be inducing reflection and scattering of the acoustic signal at greater depths, where the hydrophone is located.

Because the considered data are obtained from a hydrophone oriented in the west direction (towards the Indian ocean), one would expect very little influence on distant deep sources in a predominantly west or southwesterly direction as their ray paths sample mostly the non-shallow ocean far removed from the coast, only passing through the Leeuwin Current at the last stage. But for surface-originating sounds from other directions there may well be an impact: where the sea bottom slopes through the SOFAR channel, particularly if it does so steeply, surface sounds and seismic waves can be reflected or refracted into the SOFAR channel. Note that on account of the lower speed of sound in water compared to rock, refraction is towards the normal for seismic waves coupling into water so such coupling is not efficient for a mostly horizontal sea bottom: the hydrophone array will predominantly see seismic waves that impinge close to vertically from below (have a small slowness, high apparent velocity across the array) which are subsequently scattered by the wavy sea surface instead of propagating coherently onwards. Hence steep slopes couple better.

The detailed analysis of directional acoustic propagation is beyond the scope of the current paper and may be analysed later elsewhere.

Finally, we analyse the scaling properties of the deseasonalized fluctuations of the broadband SPL to identify the type of noise present in this dynamical system. When the noise is white, the DFA scaling exponent has value 0.5, whereas red noise has values of the exponent higher than 0.5 (Peng et al., 1994), with nonstationary red noise having exponent higher than 1 (random walk has exponent 1.5). The scaling exponent is estimating by fitting the fluctuation curve $F(s) \sim s^{\alpha}$ in a log–log plot, as shown in Fig. 7. The variable "$s$" is the scale size of the DFA, which is the size of the varying window where the fluctuations are estimated. For more details on the DFA method, we refer the reader to Kantelhardt et al. (2002).

When we apply the scaling analysis to the deseasonalized broadband SPL, in both short and long temporal range it has a high exponent (about 0.9), which means that the acoustic fluctuations are stationary red noise, and this is how they

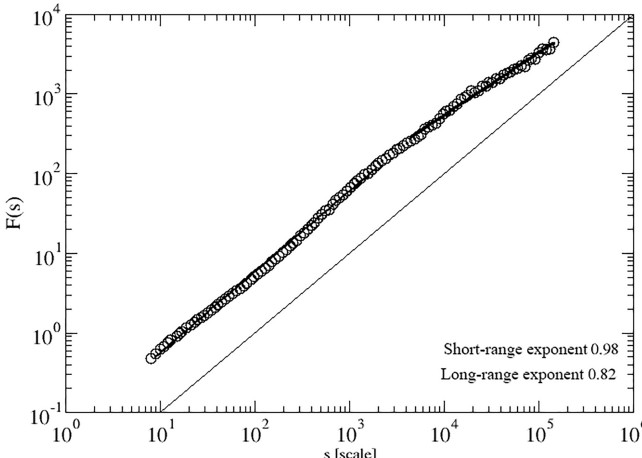

**Figure 7.** Detrended fluctuation analysis scaling curve of the broadband SPL, with estimated scaling exponent values. The straight line denotes the slope with scaling exponent 1. The scaling exponents of the curve (short term and long term) are much higher than 0.5, hence the noise is not white but red (presence of correlations); the exponent is slightly smaller than 1, which means that the fluctuations are stationary (unlike a nonstationary random walk).

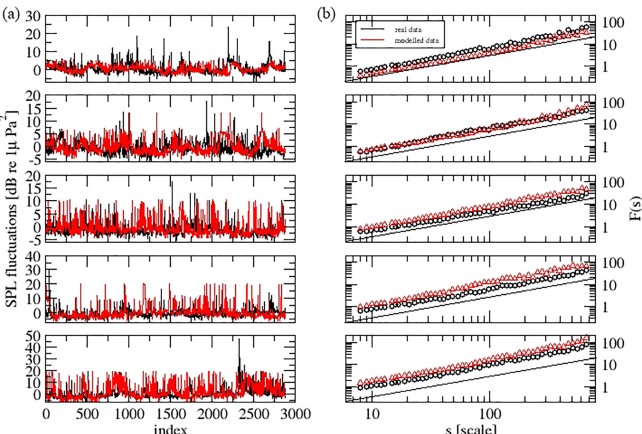

**Figure 8.** Five samples of SPL data (real, black; modelled, red). **(a)** Samples of data; **(b)** DFA scaling curves. The thin line in the panels of the right column denotes the slope with scaling exponent 1, to which the curves are very close. The modelled data have the same probability density and scaling properties as the real data.

should be modelled to represent accurately the stochastic term in Eq. (1).

It is important to model the climatic variables with colour noise rather than with basic white noise, especially when a system like this exhibits highly correlated long-term persistence (as estimated by DFA in Fig. 7 with $\alpha = 0.96$). The pattern of such fluctuations differs significantly from the pattern of the white noise: the persistent data ("with memory") is likely to have positive fluctuations tomorrow if today the fluctuations are positive. The scaling methods, such as DFA, allow one to quantify this effect, and detect the changes in the data that are not visible in time series by a naked eye, as it is illustrated in Fig. 4. If one uses white noise for modelling such complex data, the ability to analyse such data and forecast the system dynamics would be much reduced, with poor skill.

## 5 Model

Acoustic noise at the depth of 1 km can be influenced by multiple factors of natural and anthropogenic origin, and we investigate some of the possible components that could represent the dynamics of the acoustic system. While there may be various equivalent models reproducing the observed time series, we choose the simple stochastic dynamic system which generates simulated time series with very close statistical properties.

Based on the above analysis, we can formulate a stochastic model for the acoustic oceanic noise. We adopt an additive

model with the following terms:

$$\dot{z} = -U'(z, t) + T(t) + P(t) + \Phi, \quad (11)$$

where $U(t)$ is the system potential, $T(t)$ is a long-term linear trend, $P(t)$ is a seasonal trend, and $\Phi$ are red-noise fluctuations. In Eq. (11), we use time as the main variable of the time series, assuming that only the shape of the potential $U$ is defined by the state variable $z$ as described above. The parameters of the model (the global trend slope, the amplitude of the seasonality, the coefficients of the potential, the scaling exponent of the fluctuations, and the dynamics of its autocorrelation and variance) can be derived from the data and used for simulating artificial data for comparison. Such stochastically modelled artificial data can be used for a long-term forecast of acoustic data and for testing various hypotheses of the hydroacoustic dynamical system. For simplicity, we illustrate this with a triple-well-potential term $U(t)$, and further parameters derived from the broadband SPL. We show five subsamples of the SPL broadband data in Fig. 8, where observed and modelled time series are compared (left column), as well as their fluctuation curves (right column).

The model (11) is a version of the non-autonomous Langevin equation, which was previously used in analysis of paleodata; see for example Ditlevsen et al. (2005), although paleodata have different temporal resolution and patterns. This illustrates the flexibility of the stochastic modelling approach and its general applicability.

## 6 Conclusions

We have applied tipping point analysis and identified deterministic and stochastic components of the ocean acoustic data. We have discovered a possible signature of El Niño in

the deep-ocean acoustic data, which is an interesting observation confirmed by both potential analysis and direct estimation of the probability density function of the broadband SPL. Given that the hydrophones are located at depth, and the number of factors influencing the hydroacoustic system in conjunction with the global climate system is large, the investigation of the transitional mechanisms between the surface multiannual phenomena and deep-water acoustic processes may be a subject of a separate paper. The current dynamics of the acoustic fluctuations, which demonstrate slow but steady changes in early-warning indicators, gives indication of an upcoming tipping point in this hydroacoustic system, with possible appearance/disappearance of system states, which in this context denote higher/lower SPL fluctuations. Because Cape Leeuwin is a busy shipping junction in world trade, and as trading processes intensify (at the same time requiring more modern ships, with more efficient and less noisy engines), we hypothesize that frequency ranges of the oceanic acoustic noise will be affected unequally, due to multiple factors related to anthropogenic activities.

In Sardeshmukh et al. (2015), the authors consider extreme weather statistics and warn against seeking anthropogenic components in data with heavy-tailed non-Gaussian distributions. In agreement with their opinion, we do not attempt to introduce an anthropogenic term in our model. However, the acoustic impact of shipping and other anthropogenic factors on the marine wildlife have already been reported: anthropogenic sounds affect vocalizing baleen whales, with distance of impact up to 200 km (Risch et al., 2012); such disturbances cause behavioural changes in large animals, and consequently reduce their acoustic presence in the area of observation.

Some frequency bands may decrease in level because of the technological changes: new developments in quieter engine technology, establishment of noise mitigation standards, and renewal of the fleets. In particular, there is global large-scale replacement of heavy-tonnage ships, where in some categories, like "cargo", "containers" and "bulk carriers", ships of age above 25 years are no longer present; see the report of the European Maritime Safety Agency (EMSA, 2015).

Other potential causes of trends in ocean noise levels include changes in the frequency of other anthropogenic sources such as geophysical surveying, changes in the number and distributions of biological sources such as large cetaceans, changes in natural sources of sound such ice breaking and ice formation, and changes in the ocean environment which may affect the propagation of sound (for example, sea temperature).

By taking into account all the components of the proposed model (11), i.e. the global trend, the seasonal trend, the asymmetric system potential structure, and the long-term correlated red noise, one can reproduce the considered acoustic data. Our analysis allows one to understand these main components and derive their specific parameters, which are then used to forecast data, and thus validate our understanding of this dynamical system. Indeed, there may be other models that can produce similar structure of time series. The advantage of our model is its simplicity and adequate representation of the main geophysical processes of this dynamical system.

The hypothesis of the possible influence of El Niño appeared in the course of our research and was unexpected. Therefore, our modelling approach, in principle, is capable of discovering such interesting signatures in the data for further investigation. This demonstrates the capability of the proposed data analysis, on its part, to stimulate geophysical research.

*Data availability.* . TS3

*Competing interests.* The authors declare that they have no conflict of interest. TS4

*Acknowledgements.* The views expressed herein are not the opinion of the CTBTO. Valerie N. Livina, Peter Harris, Kostas Sotirakopoulos, Lian Wang, and Stephen Robinson are funded by the National Measurement Office. The authors are grateful to the specialists of the virtual Data Exploitation Centre (vDEC) for providing the CTBTO data in their servers.

Edited by: Stéphane Vannitsem
Reviewed by: two anonymous referees

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

**Remarks from the typesetter**

TS1    The composition of all figures has been adjusted to our standards.

TS2    Copernicus Publications collects the DOIs of data sets, videos, samples, model code, and other supplementary/underlying material or resources as well as additional outputs. These assets should be added to the reference list (author(s), title, DOI, and year) and properly cited in the article. If no DOI can be registered, assets can be linked through persistent URLs. This is not seen as best practice and the persistence of the URL must be secured.

TS3    Please provide a statement on how your underlying research data can be accessed. If the data are not publicly accessible, a detailed explanation of why this is the case is required. The best way to provide access to data is by depositing them (as well as related metadata) in reliable public data repositories, assigning digital object identifiers (DOIs), and properly citing data sets as individual contributions. Please indicate if different data sets are deposited in different repositories or if data from a third party were used. If no DOI is available, assets can be linked through persistent URLs to the data set itself (not to the repositories' home page). This is not seen as best practice and the persistence of the URL must be secured.

TS4    Declaration of all potential conflicts of interest is required by us as this is an integral aspect of a transparent record of scientific work. If there are possible conflicts of interest (see http://publications.copernicus.org/services/competing_interests_policy.html), please state what competing interests are relevant to your work.

TS5    Please list all authors (name, initial).

TS6    Please provide article number with DOI or page range.

TS7    Please provide DOI or page range.

TS8    Please provide DOI or page range.

TS9    Please list all authors (name, initial).

TS10    Please list all authors (name, initial).

TS11    Please provide article number with DOI or page range.

TS12    Please provide DOI or page range.

TS13    Please provide last access date.

TS14    Please provide article number with DOI or page range.

TS15    Please provide article number with DOI or page range.

TS16    Please provide article number with DOI or page range.