# Peer review of "Tipping point analysis of ocean acoustic noise"

_Nonlinear Processes in Geophysics, 2017_

## Referee Comment (RC1) · Anonymous Referee #1 · 5 Oct 2017

This paper analyzes a 14-yr long acoustic time series by estimating its non-stationary and stationary statistical characteristics: seasonal cycles and long-term trends in five different frequency bands, lag-1 autocorrelation and variance in large sliding windows of sizes 3-9 years, as well as estimates of the potential function in the sliding windows of sizes from a few days to a year. The authors find statistically significant long-term trends in the acoustic system dynamics potentially attributable to anthropogenic influences on climate, and argue for a possibility of a future bifurcation based on the tipping-point analysis. On shorter time scales, the authors argue for a connection between the structure of the system's underlying potential and ENSO variability. A stochastic model is developed that is able to mimic diverse statistical properties of the observed acoustic data.

The paper is well written in general, but needs a few clarifications in some places, in particular with regards to the conclusions about the connection with ENSO and the

formulation of the stochastic model. The results (especially Figs. 4 and 5) look interesting, but I feel that interpretation of these results using buzzwords (anthro, ENSO) is somewhat arbitrary, and needs a broader discussion. In the same way, a few words about possible alternatives to the stochastic formulation the authors developed would also be in order.

Detailed comments:

(1) Fig. 4: The record is only 14-yr long, and there is definitely a lot of internal variability in the climate system that can contribute to decadal trends on top of any possible anthropogenic effects.

(2) Fig. 5: The text claims a connection with ENSO, but it's not immediately obvious from comparing the top and bottom panels of the figure. Could the authors produce a bit more quantitative measure of this association (for example, plotting a 1-D time series of the color plot and correlating it with ENSO indices)?

(3) Fig. 6: Such skewed distributions can be generated by the stochastic models with nonlinear deterministic part driven by the additive noise, but also with linear models driven by multiplicative noise. See Compo et al. (2015) and references therein (http://journals.ametsoc.org/doi/abs/10.1175/JCLI-D-15-0020.1). So the choice of the appropriate stochastic model may not be that obvious.

(4) Section 5, eq. (11). I am confused about the model. Should it be z-dot (not z-prime) on the left-hand side, as in (4)? But then are $T(t)$ and $P(t)$ added on top of the potential model driven by the red-noise (Phi)? In this case, they cannot be the part of Eq. (11). ?? Do the coefficients of the potential continuously depend on time (as implied by Fig. 5)? Or is the model trained on subsamples of data with a given potential structure (e.g., two-well, three-well)? Is it really a surprise that the model reproduces the observed statistics? This section needs to be clarified.

2017-48, 2017.

---

## Referee Comment (RC2) · Anonymous Referee #2 · 6 Nov 2017

The paper "Tipping point analysis of ocean acoustic noise", by Valerie Livina et al. is an interesting study of the ocean acoustic dynamical system analysed in terms of its deterministic and stochastic components. Using tipping point analysis, they detect a few bifurcations in the acoustic data measured off the south-west shore of Australia, the strongest of which surprisingly coincides with the recent (2016) very strong El Niño event. This result suggests that the signature of El Niño can be found in the distant ocean acoustic data, whilst it may inspire future studies to understand the origin of other perturbations which the proposed method detects. The paper is well-written and of interest to the geophysics community, as well as to specialists in other disciplines: acoustics, time series analysis, statistical physics, and others.

One can see the signature of the recent (2016) very strong El Niño in the potential analysis plot, Fig 5, which supports their hypothesis. Do the authors suggest using acoustic signal as a possible precursor of El Niño? Will this be useful in detection of El

Niño in future?

The lower panel of Fig 4, shows a long-term decreasing trend in variance, which seems to stabilise shortly before 2015. There is a sharp increase in autocorrelations around the time when the variance seems to stabilise. Do the authors attribute this to El Niño, too?

Although the clearest change in the number of potential wells (Fig 5, page 10) coincides with the strongest El Niño in 2016, a few less clear changes in potential structure are visible in the upper panel of Fig 5 which apparently do not coincide with El Niño events (lower panel). It may be useful to signal the El Niño events more clearly. I have a few suggestions regarding Figure 5, page 10:

- The x-axes of the upper and lower panels are not well aligned. This makes it harder to notice the coincidence between the 2016 El Niño event (lower panel), and the change from triple to double potential wells (upper panel). To make the Figure clearer, the scale could be aligned, and a gridline added in the lower panel.

- should the y-axis of the lower panel be aligned to make the zero value for the ONI and SOI coincide?

- The SOI index looks very noisy. If El Niño is indicated by simultaneous positive ONI and negative SOI values, the only very clear indication of El Niño is the last one (2016). Accordingly, this is when we see the clearest change in the number of potential wells, spanning all time scales (upper panel). Why do the authors use these noisy indices instead of direct records of, for example, sea-surface temperature?

- Is it important that the changes in the number of potential wells span the entire time scale at a given time? Regarding the results shown on Figure 7, page 12, could the authors explain what is the variable "s"?

The authors could explain better why using colour noise is important in modelling climatic variables.

Minor comments/typos:

Page 2, line 55, is it "polynomial of given order", instead of "even order"?

Page 4, line 100: "(. . .) and sigma is the noise level." Replace eta by sigma.

Page 7, line 164: what is n in "r=n-d"? In Eq 10, do both t and k run from 1 to m?

Fig 7, page 12: could resolution be improved?

The authors should correct the Eq 11, page 13, where the derivative in time should be denoted by dot. Fig 8, page 14: I have difficulty reading the legend in the upper right panel. The font (or figure) is too small and the resolution is not optimal.

---

## Referee Comment (RC3) · Anonymous Referee #1 · 14 Nov 2017

I thank the authors for their responses to my original comments. I believe this paper is suitable for publication in NPG.

---

## Author Comment (AC1) · 14 Nov 2017

**Response to Reviewer#1's comments**

on paper "Tipping point analysis of ocean acoustic noise"

by V. N. Livina, A. Brouwer, P. Harris, L. Wang, K. Sotirakopoulos, S. Robinson

We would like to thank the Reviewer for the helpful comments on the paper. We have prepared a marked manuscript with added text as described below.

- **Comment 1.** Fig. 4: The record is only 14-yr long, and there is definitely a lot of internal variability in the climate system that can contribute to decadal trends on top of any possible anthropogenic effects.
  - **Response.** We agree with the reviewer that there may be various causes for the trend, and antropogenic influence may be only a part of contribution on the background of the various climatic factors this is why, in particular, we investigate the connection with El Niño events.

Acoustic noise at the depth of one kilometre can be influenced by multiple factors of natural and anthropogenic origin, and we discuss some of the possible components that could represent the dynamics of the acoustic system. While there may be various equivalent models reproducing the observed time series, we choose the simple stochastic dynamic system which generates simulated time series with very close statistical properties.

It has already been reported that antropogenic acoustic signals affect vocalizing baleen whales, with distance of impact up to 200 km [Risch et al 2012]: such disturbances cause behavioural changes in large animals, and consequently reduce their acoustic presence in the area of observation. This may be one of the reasons of declining trend in the ocean acoustic noise in particular frequency ranges.

We have added this paragraph into the manuscript.

- **Comment 2.** Fig. 5: The text claims a connection with ENSO, but its not immediately obvious from comparing the top and bottom panels of the figure. Could the authors produce a bit more quantitative measure of this association (for example, plotting a 1-D time series of the color plot and correlating it with ENSO indices)?
  - **Response.** We have attempted to generate a time series from the coloured potential plot at a chosen time scale (0.5 yr). Note that the values of such time series can only be integer numbers, as the variable is the number of the potential wells in the dynamical system. On the other hand, the El Niño indices are continuous variables, which denote an event when they are varying below or above certain level. To make them comparable with the extracted potential metrics described above, we produce a binary variable for the ONI index time series: such variable

is 1 when ONI is bigger than 0.2 (different sources suggest different threshold, sometimes 0.5, sometime 0 as indication of El Niño events). The varying number of wells in the potential plot was converted into a binary index, which equals 1 when there is "non-regular" (not triple-well-potential) state of the dynamic system. Accordingly, we have prepared a new version of Fig.5, with three panels, where this is illustrated.

For several known El-Nino events (2003, 2004, 2006, 2009, 2015-2016), there is agreement between the two binary indices. The potential index has several additional oscillations which might have been caused by different factors among many that affect this complex dynamic system (yet another possible cause may be La Niña events, when ONI index is negative — see the new version of Figure 5).

We have added this text in the manuscript.

- Comment 3. Fig. 6: Such skewed distributions can be generated by the stochastic models with nonlinear deterministic part driven by the additive noise, but also with linear models driven by multiplicative noise. See Compo et al. (2015) and references therein (http://journals.ametsoc.org/doi/abs/10.1175/JCLI-D-15-0020.1). So the choice of the appropriate stochastic model may not be that obvious.
  - **Response.** As we understand it, the reviewer mentions the paper by Sardeshmukh et al, JoC 2015. The paper considers extreme weather statistics and warns against seeking anthropogenic components in data with heavy-tailed non-Gaussian distributions. We agree with this statement, and we do not attempt to introduce an anthropogenic term in our model. However, the acoustic impact of shipping and other anthropogenic factors on the marine wildlife has been reported (as in the reference above and those cited in the paper), and we add this as a discussion topic rather than a modelling component.

We have added this reference and a few sentences on this into the manuscript.

- Comment 4. Section 5, eq. (11). I am confused about the model. Should it be z-dot (not zprime) on the left-hand side, as in (4)? But then are T(t) and P(t) added on top of the potential model driven by the red-noise (Phi)? In this case, they cannot be the part of Eq. (11)? Do the coefficients of the potential continuously depend on time (as implied by Fig. 5)? Or is the model trained on subsamples of data with a given potential structure (e.g., two-well, three-well)?
  - **Response.** The reviewer is obviously right about the typo in the left-hand-side of Eq.11, it is derivative in t and should be  $\dot{z}$ . We have corrected this.

The potential term and the stochastic noise in the model equation are used to reproduce the fluctuations that are obtained after subtraction of seasonal and long-term linear trend. The data that is modelled with the potential forecasting technique is shown in Figure 1, right panels. These fluctuations also include the red noise component. Its characteristics, from our point of view, do vary with time, as Figure 4 shows.

The original data (Figure 1, left panels) can be reproduced by combining all the components of model (11), which is a version of the nonautonomous Langevin equation. We follow this modelling approach as it was used in paleostudies, see for example [Ditlevsen et al, JoC 2005], although paleodata has different temporal resolution and patterns. This illustrates the flexibility of the stochastic modelling approach and its general applicability.

We have added this paragraph into the paper.

- **Comment 5.** Is it really a surprise that the model reproduces the observed statistics? This section needs to be clarified.
  - **Response.** In our opinion, only by taking into account all the components of the proposed model, i.e. the global trend, the seasonal trend, the asymmetric system potential structure, and the long-term correlated red noise, one can reproduce the considered acoustic data. Our analysis allows one to understand these main components and derive their specific parameters, which are then used to forecast data, and thus validate our understanding and the proposed stochastic model. Indeed, there may be other models that can produce similar structure of time series. The advantage of our model is its simplicity and adequate representation of the main geophysical processes of this dynamical system.

The hypothesis of the possible influence of El Niño appeared in the course of our research and was unexpected. Therefore, our modelling approach, in principle, is capable of discovering such interesting signatures in the data for further investigation. This demonstrates the capability of the proposed data analysis, on its part, to stimulate geophysical research.

We have added this text into the paper.

We hope that the revised manuscript is now suitable for publication in the Nonlinear Processes in Geophysics.

Yours sincerely,

V. N. Livina, A. Brouwer, P. Harris, L. Wang, K. Sotirakopoulos, and S. Robinson

---

## Author Comment (AC3) · 14 Nov 2017

**Response to Reviewer #2's comments**

on paper "Tipping point analysis of ocean acoustic noise"

by V. N. Livina, A. Brouwer, P. Harris, L. Wang, K. Sotirakopoulos, S. Robinson

We would like to thank the Reviewer for the helpful comments on the paper. We have prepared a marked manuscript with added text as described below.

- **Comment 1.** One can see the signature of the recent (2016) very strong El Niño in the potential analysis plot, Fig 5, which supports their hypothesis. Do the authors suggest using acoustic signal as a possible precursor of El Niño? Will this be useful in detection of El Niño in future?
  - **Response.** We do not claim that the potential colour plot could be used for early warning signals, in this system or in others. Moreover, there may be other factors causing structural changes in the acoustic data, rather than El Niño or La Niña. On the other hand, detection of such changes, indeed, can be useful for other studies that could investigate attribution of structural variability, and here the technique of potential analysis might be very useful.

We have added this into the revised manuscript.

- **Comment 2.** The lower panel of Fig 4, shows a long-term decreasing trend in variance, which seems to stabilise shortly before 2015. There is a sharp increase in autocorrelations around the time when the variance seems to stabilise. Do the authors attribute this to El Niño, too?
  - **Response.** In our opinion, the noticeable change in these indicators is related to the unusually large El Niño event of 2015-2016. One can see that the variance decline slows down and autocorrelation sharply rises, which means that the increase in memory is not accompanied by increasing amplitude of acoustic fluctuations. Such effects may happen when a dynamical system experiences critical slowing down prior to a bifurcational tipping. As we hypothesise that the El Niño signature may be related to changes in both oceanic dynamics and fauna, the increasing memory in the acoustic data may reflect, for instance, the observation that during the El Niño the Cape Leeuwin current slows down [Feng et al 2003]. Such slower ocean dynamics introduces more inertia in the dynamical system, and therefore higher temporal memory/autocorrelations.

We have added this text in the manuscript.

**Comment 3.** Although the clearest change in the number of potential wells (Fig 5, page 10) coincides with the strongest El Niño in 2016, a few less clear changes in potential structure are visible in the upper panel of Fig 5 which apparently do not coincide

with El Niño events (lower panel). It may be useful to signal the El Niño events more clearly. I have a few suggestions regarding Figure 5. The x-axes of the upper and lower panels are not well aligned. This makes it harder to notice the coincidence between the 2016 El Niño event (lower panel), and the change from triple to double potential wells (upper panel). To make the Figure clearer, the scale could be aligned, and a gridline added in the lower panel.

- **Response.** We have prepared a new version of Figure 5, with three panels, and aligned the scales of the panels more accurately. We calculated, for easier comparison, two binary indices derived from the ONI index and from the single level of the potential plot at the scale 0.5 yr. The bars in the bottom panel show the occurence of El Niño events in the ONI index (which is less noisy than SOI), and at the same time we plot a binary index showing periods when the system potential does not follow its "regular" three-well-potential pattern; these two indices have agreement in several periods corresponding to the known El Niño events (2003, 2004, 2006, 2009, 2015-2016), which illustrates our hypothesis of the El Niño signature in the acoustic data.
- **Comment 4.** Should the y-axis of the lower panel be aligned to make the zero value for the ONI and SOI coincide?
  - **Response.** We have corrected this.
- **Comment 5.** Is it important that the changes in the number of potential wells span the entire time scale at a given time?
  - **Response.** The vertical span of the features of the potential plot (the specks of different colours) correspond to the time scale of the change, i.e. the size of the time window, within which the change has been detected. As El Niño is a seasonal phenomenon (except the unusually long event of 2015-2016), most of such specks are located within the window of size one year. The large event of 2015-2016, indeed, extends higher than that. To address this time scale, we derived the binary potential index using the detection data at fixed time scale of 0.5 year, at which most El Niño events should be present in the detection statistics.

We have added this text into the manuscript.

- **Comment 6.** Regarding the results shown on Figure 7, page 12, could the authors explain what is the variable "s"?
  - **Response.** The variable 's' is the scale size of the DFA, which is the size of the varying window where the fluctuations are estimated. For more details on the DFA method, we refer the reader to [Kantelhardt et al 2002].

- **Comment 7.** The authors could explain better why using colour noise is important in modelling climatic variables.
  - **Response.** It is important to model the climatic variables with colour noise rather than with basic white noise, especially when a system like this exhibits highly correlated long-term persistence (as estimated by DFA in Fig.7 with  $\alpha = 0.96$ ). The pattern of such fluctuations differs significantly from the pattern of the white noise: the persistent data ('with memory') is likely to have positive fluctuations tomorrow if today the fluctuations are positive. The scaling methods, such as DFA, allow one to quantify this effect, and detect the changes in the data that are not visible in time series by a naked eye, as it is illustrated in Fig.4. If one uses white noise for modelling such complex data, the ability to analyse such data and forecast the system dynamics would be much reduced, with poor skill.

We have added this text into the manuscript.

- **Comment 8.** Page 2, line 55, is it "polynomial of given order", instead of "even order"?
  - **Response.** We use the expression "even order" to refer to the polynomials of order 2n that model n number of wells. In the case of one well (symmetric), it is a polynomial of order 2; in the case of two wells, it is a polynomial of order 4, etc.
- Comment 9. Page 4, line 100: "(: : :) and sigma is the noise level." Replace eta by sigma.

**Response.** We have corrected this.

Comment 10. Page 7, line 164: what is n in "r=n-d"? In Eq 10, do both t and k run from 1 to m?

**Response.** We have corrected this.

- **Comment 11.** Fig 7, page 12: could resolution be improved?
  - **Response.** We have a good quality (large-size) eps-file of this figure prepared, but for the lighter version of the pdf-file in the current form we used a low-resolution jpg-figure, to make the manuscript pdf-file lighter.
- Comment 12. The authors should correct the Eq 11, page 13, where the derivative in time should be denoted by dot.

**Response.** We have corrected this.

**Comment 13.** Fig 8, page 14: I have difficulty reading the legend in the upper right panel. The font (or figure) is too small and the resolution is not optimal.

**Response.** Similarly to the above comment, these are low-resolution technical jpg-files, which are used to produce a small-size pdf-file for easy download at reviewing stage. We are prepared to provide all the figures in large heavy eps-files if necessary.

We hope that the revised manuscript is now suitable for publication in the Chaos magazine.

Yours sincerely, V. N. Livina, A. Brouwer, P. Harris, L. Wang, K. Sotirakopoulos, and S. Robinson

---

## Referee Comment (RC4) · Anonymous Referee #2 · 17 Nov 2017

The authors' reply are adequate and useful, and the paper is clearly improved. I find it suitable for publication.

———————————————